# Umbilical Cord Mesenchymal Stem Cell-Derived Extracellular Vesicles Enhance Chondrocyte Function by Reducing Oxidative Stress in Chondrocytes

**DOI:** 10.3390/ijms26167683

**Published:** 2025-08-08

**Authors:** Che-Wei Wu, Yao-Hui Huang, Pei-Lin Shao, Ling-Hua Chang, Cheng-Chang Lu, Chung-Hwan Chen, Yin-Chih Fu, Mei-Ling Ho, Je-Ken Chang, Shun-Cheng Wu

**Affiliations:** 1Regenerative Medicine and Cell Therapy Research Center, Kaohsiung Medical University, No. 100, Shih-Chuan 1st Road, Kaohsiung 807, Taiwan; tkdiven@gmail.com (C.-W.W.); justin23523@gmail.com (Y.-H.H.); linghua_chang@yahoo.com.tw (L.-H.C.); cclu@kmu.edu.tw (C.-C.L.); hwan@kmu.edu.tw (C.-H.C.); microfu@gmail.com (Y.-C.F.); homelin@kmu.edu.tw (M.-L.H.); jkchang@cc.kmu.edu.tw (J.-K.C.); 2Orthopaedic Research Center, College of Medicine, Kaohsiung Medical University, Kaohsiung 807, Taiwan; 3Department of Food Science, Yuanpei University of Medical Technology, Hsinchu City 30015, Taiwan; 4Department of Nursing, Asia University, Taichung 413, Taiwan; m8951016@gmail.com; 5Department of Orthopedics, Kaohsiung Municipal Siaogang Hospital, Kaohsiung Medical University, Kaohsiung 812, Taiwan; 6Department of Orthopedics, Kaohsiung Medical University Hospital, Kaohsiung Medical University, Kaohsiung 807, Taiwan; 7Department of Orthopedics, College of Medicine, Kaohsiung Medical University, Kaohsiung 807, Taiwan; 8Ph.D. Program in Biomedical Engineering, College of Medicine, Kaohsiung Medical University, Kaohsiung 807, Taiwan

**Keywords:** articular cartilage tissue engineering (ACTE), chondrocyte function, oxidative stress, umbilical cord mesenchymal stem cells (UCMSCs), extracellular vesicles (EVs), miRNA

## Abstract

Articular cartilage (AC) has a very limited capacity for self-healing once damaged. Chondrocytes maintain AC homeostasis and are key cells in AC tissue engineering (ACTE). However, chondrocytes lose their function due to oxidative stress. Umbilical cord mesenchymal stem cells (UCMSCs) are investigated as an alternative cell source for ACTE. MSCs are known to regulate tissue regeneration through host cell modulation, largely via extracellular vesicle (EV)-mediated cell-to-cell communication. The purpose of this study was to verify whether UCMSC-derived EVs (UCMSC-EVs) enhance chondrocyte function. The mean particle sizes of the UCMSC-EVs were 79.8 ± 19.05 nm. Transmission electron microscopy (TEM) revealed that UCMSC-EVs exhibited a spherical morphology. The presence of CD9, CD63, and CD81 confirmed the identity of UCMSC-EVs, with α-tubulin undetected. UCMSC-EVs maintained chondrocyte survival, and increased chondrocyte proliferation after intake by chondrocytes. UCMSC-EVs upregulated mRNA levels of SOX-9, collagen type II (Col-II), and Aggrecan, while decreasing collagen type I (Col-I) levels. UCMSC-EVs reduced the oxidative stress of chondrocytes by reducing mitochondrial superoxide production and increasing protein levels of SOD-2 and Sirt-3 in chondrocytes. The 50 most abundant known microRNAs (miRNAs) derived from UCMSC-EVs were selected for gene ontology (GO) and Kyoto Encyclopedia of Genes and Genomes (KEGG) pathway analyses. GO analysis revealed enrichment in pathways associated with small GTPase-mediated signal transduction, GTPase regulatory activity, and mitochondrial matrix. The KEGG analysis indicated that these miRNAs may regulate chondrocyte function through the PI3K-Akt, MAPK, and cAMP signaling pathways. In summary, this study shows that UCMSC-EVs enhance chondrocyte function and may be applied to ACTE.

## 1. Introduction

Articular cartilage has a very limited capacity for self-healing once damaged, making the successful repair of articular cartilage defects a major challenge in orthopedics [1,2,3]. These articular cartilage defects are recognized as one of the primary causes of osteoarthritis (OA), which is the most prevalent joint disorder worldwide [3,4,5,6]. Current clinical treatments, such as subchondral drilling and microfracture, have been widely applied to manage articular cartilage defects [1,2,3]. However, these methods are limited by the formation of fibrocartilage in the defect area [1,2,3]. Unlike native hyaline cartilage, fibrocartilage possesses inferior mechanical properties, cannot adequately perform the biomechanical functions of hyaline cartilage, and is prone to early degeneration [3,7,8,9]. To date, no clinical strategy has been able to consistently regenerate native hyaline cartilage in articular cartilage defects, and effective cartilage repair remains an unmet clinical need. In response to these limitations, articular cartilage tissue engineering (ACTE) has emerged as an alternative approach [3].

ACTE relies on three fundamental components, cellular elements, a biocompatible scaffold, and bioactive molecules, to promote cartilage regeneration [10]. Native articular cartilage consists of hyaline cartilage containing a sparse population of chondrocytes, which are responsible for maintaining cartilage homeostasis [4]. Therefore, chondrocytes have been extensively studied as a cell source for ACTE [4]. To obtain a sufficient number of cells for clinical use, in vitro cell expansion of chondrocytes is commonly performed [11,12,13]. However, this process leads to a progressive loss of chondrocyte function, impairing their ability to produce hyaline cartilage matrix [6,14]. This functional decline is believed to result from “stress responses” induced by in vitro culture conditions [14]. Multiple investigations have reported that elevated oxidative stress—especially excessive production of reactive oxygen species (ROS)—is a key factor underlying chondrocyte dysfunction during in vitro expansion [14,15,16]. Antioxidant treatments, including *N*-acetyl cysteine (NAC) and ascorbic acid, have demonstrated efficacy in alleviating ROS-induced cellular damage and in delaying the functional deterioration of chondrocytes during culture [17,18]. Nevertheless, developing an effective and controllable delivery system for sustained antioxidant release within articular cartilage defects remains a significant challenge. Therefore, restoring the functional capacity of chondrocytes after in vitro expansion continues to be a major obstacle in chondrocyte-based ACTE.

In ACTE, mesenchymal stem cells (MSCs) are widely regarded as a promising alternative cell source, given their capacity to undergo chondrogenic differentiation under appropriate induction conditions [19]. MSCs were sourced from various adult (e.g., adipose, bone marrow, synovium) and fetal (e.g., umbilical cord, placenta, amniotic fluid) tissues [20,21]. The umbilical cord (UC) is recognized as a highly favorable source of MSCs owing to its non-invasive collection process and ready availability, which circumvent any harm to either the mother or the infant [22]. Although MSCs share common surface markers—CD105, CD73, and CD90—they display distinct functional characteristics depending on their tissue source [23,24]. Compared to adult-derived MSCs, UCMSCs exhibit more primitive biological characteristics, reduced susceptibility to genetic mutations and viral contamination, and an enhanced proliferation capacity under in vitro conditions [22]. However, MSC-based therapy is limited by cell heterogeneity, immune responses, and risk of tumor formation or disease transmission [25,26]. Moreover, the storage and transportation of these cells require stringent conditions to preserve their viability and functional integrity [26]. This limits the feasibility of directly replacing chondrocytes with MSCs.

Recently, it became evident that transplanted MSCs do not necessarily need to migrate into the damaged articular cartilage and differentiate into chondrocytes [3]. MSCs are recognized to coordinate tissue regeneration via modulating the cell functions of the host tissue, e.g., immunomodulatory effects [27]. MSCs exert therapeutic effects largely through the paracrine signaling of secretome, which consists of extracellular vesicles (EVs) and soluble factors such as peptides and cytokines [28]. EVs are membrane-bound particles released from cells that carry biological messages, such as proteins, mRNAs, and miRNAs, to target cells, modulating their genotype and phenotype [29,30]. Recent studies indicate that EVs are capable of mimicking the numerous therapeutic functions of MSCs, offering a promising acellular approach for cartilage regeneration [3,29]. Compared to live cell therapies, EV-based treatments provide notable benefits, such as reduced immunogenicity and enhanced stability during storage and transport [31]. In recent years, studies have highlighted the beneficial effects of UCMSC-derived EVs (UCMSC-EVs) on articular cartilage, demonstrating their anti-inflammatory properties, ability to reduce cartilage degradation, and capacity to attenuate chondrocyte senescence [31,32,33,34,35]. Therefore, UCMSC-EVs may serve as a therapeutic adjunct for restoring chondrocyte function in chondrocyte-based ACTE. Furthermore, despite evidence from in vivo studies supporting the therapeutic potential of UCMSC-EVs, there remains a lack of focused cellular-level investigations, especially concerning the direct interactions between UCMSC-EVs and chondrocyte function.

In the present study, we hypothesized that UCMSC-EVs would promote chondrocyte function. To test this hypothesis, we isolated and characterized UCMSC-EVs and evaluated their effects on various aspects of chondrocyte function, including cell survival, proliferation, mRNA levels of SOX-9, collagen type II (Col-II), Aggrecan, Col-I, and oxidative stress. Additionally, we analyzed the microRNA (miRNA) profile of UCMSC-EVs to explore their potential molecular mechanisms of action.

## 2. Results

### 2.1. Characterization of UCMSCs and UCMSC-EVs

UCMSCs exhibited a spindle-like morphology upon adhesion to the plastic culture plate (Figure 1A). Flow cytometry was used to identify the surface markers of UCMSCs; the cultured UCMSCs exhibit high expression of CD105, CD90, and CD73, and low expression of CD34, CD45, and HLA-DR (Figure 1B). Immediately after UCMSC-EVs were isolated, the particle size, particle number, protein composition, and morphology of the UCMSC-EVs were analyzed. The mean particle sizes of the UCMSC-EVs were 79.8 ± 19.05 nm. The protein composition of the UCMSC-EVs was determined through flow cytometry analysis. The isotype control was used to assess non-specific antibody binding and differentiate true signal from background noise. The result indicated that UCMSC-EVs were positive for CD9, CD63, and CD81 (Figure 1C). The ratio of UCMSC-EVs that were positive for CD9, CD63, and CD81 were 22.8%, 5.5%, and 12.1%, respectively (Figure 1C). Western blotting analysis revealed that UCMSC-EVs were positive for CD63 and CD81, but negative for α-tubulin, confirming their exosomal identity (Figure 1D). TEM further validated the UCMSC-EVs’ morphology, with captured images showing spherical structures consistent with typical EVs (Figure 1E). Together, these findings demonstrate that UCMSCs secreted EVs into the culture medium and that these vesicles were successfully isolated.

### 2.2. Visualization of UCMSC-EV Internalization in Chondrocytes

Red-fluorescence-labeled UCMSC-EVs (10^9^ particles/mL) were incubated with chondrocytes for 7 days to evaluate cellular uptake. Red-fluorescence-labeled UCMSC-EVs were not detected within chondrocytes in either the Control or UCMSC-EVs group on day 1 (Figure 2). By day 3, labeled vesicles were internalized by chondrocytes in the UCMSC-EVs group, with fluorescence signals observed in the cytoplasm and perinuclear regions (Figure 2). On day 7, a more pronounced uptake was noted, indicating a time-dependent internalization of UCMSC-EVs by chondrocytes. These findings demonstrate that chondrocytic uptake of UCMSC-EVs began after three days of incubation.

### 2.3. Effect of UCMSC-EVs on Survival and Proliferation of Chondrocytes

To evaluate the effect of UCMSC-EVs on chondrocyte survival, cells were incubated for 7 days with varying concentrations of UCMSC-EVs (10^8^–10^10^ particles/mL). Live/dead staining demonstrated that chondrocytes remained viable across all groups, with no visibly dead cells detected in either the Control or UCMSC-EV-treated groups (Figure 3A,B). In terms of proliferation, treatment with UCMSC-EVs markedly enhanced chondrocyte proliferation at concentrations of 10^9^ and 10^10^ particles/mL, as compared to the Control group (Figure 3C). Collectively, these findings indicate that UCMSC-EVs support chondrocyte survival across a broad concentration range (10^8^–10^10^ particles/mL), while specifically promoting proliferation at higher concentrations (10^9^–10^10^ particles/mL).

### 2.4. Effect of UCMSC-EVs on mRNA Expression of Chondrogenic Genes (SOX-9, Col-II, and Aggrecan) and Fribrocartilgous Gene (Col-I) of Chondrocytes

To evaluate the effect of UCMSC-EVs on the mRNA expression of chondrogenic and fibrocartilaginous markers in chondrocytes, cells were treated with UCMSC-EVs at concentrations of 10^9^–10^10^ particles/mL for 7 days. On day 7, mRNA levels of chondrogenic genes (SOX-9, Col-II, and Aggrecan) and the fibrocartilaginous marker gene (Col-I) were analyzed. The results demonstrated that UCMSC-EVs treatment upregulated the expression of chondrogenic markers while downregulating Col-I expression. Notably, the mRNA levels of SOX-9, Col-II, and Aggrecan were significantly higher in the UCMSC-EVs group than in the Control group, indicating a promotive effect on chondrogenic phenotype (Figure 4A–C). By contrast, the mRNA expression level of Col-I was lower in the UCMSC-EVs group than Control group (Figure 4D). Overall, the results revealed that UCMSC-EVs enhanced the mRNA expression of chondrogenic genes while suppressing the expression of fibrocartilaginous gene in chondrocytes.

### 2.5. Effect of UCMSC-EVs on Mitochondrial Superoxide Production and the Protein Levels of SOD-2 and Sirt-3 of Chondrocytes

We further test whether UCMSC-EVs influence the mitochondrial superoxide production and the protein levels of SOD-2 and Sirt-3 of chondrocytes. After UCMSC-EVs treatment, the level of mitochondrial superoxide production in chondrocytes was decreased (Figure 5A). In comparison to that in the Control group, the mitochondrial superoxide production in chondrocytes was decreased in the UCMSC-EVs group (10^9^–10^10^ particles/mL) (Figure 5A). We further confirmed the effect of UCMSC-EVs on the protein levels of SOD-2 and Sirt-3 in chondrocytes. We found that UCMSC-EVs also increased the protein levels of SOD-2 and Sirt-3 in chondrocytes. In comparison to that in the Control group, the expression of SOD-2 and Sirt 3 in chondrocytes was increased in the UCMSC-EV group (10^10^ particles/mL) (Figure 5B,C).

### 2.6. Bioinformatics Analysis of UCMSC-EV miRNAs

Three independent batches of UCMSC-EVs were isolated for miRNA profiling. Total RNA was extracted from each batch and used for small RNA sequencing. The 50 most abundant known miRNAs present in UCMSC-EVs were ranked according to total read count (Figure 6). To explore the potential regulatory effects of these miRNAs on chondrocyte function, GO and KEGG enrichment analyses were conducted. GO analysis revealed significant enrichment in pathways related to small GTPase-mediated signal transduction, GTPase regulatory activity, and the mitochondrial matrix, suggesting a potential role of UCMSC-EVs in modulating chondrocytic signaling and mitochondrial processes (Figure 7), whereas the KEGG analysis revealed the key roles of the phosphatidylinositol 3′-kinase (PI3K)-Akt, mitogen-activated protein kinase (MAPK), and Cyclic adenosine 3′,5′-monophosphate (cAMP) signaling pathways (Figure 8). These findings underscore the potential role of miRNAs enriched in UCMSC-EVs in modulating chondrocyte function.

## 3. Discussion

Articular cartilage repair remains a major challenge in orthopedics due to its poor self-healing capacity. Chondrocyte-based ACTE has been extensively studied for its potential to regenerate articular cartilage, but it is limited by fibrocartilage formation [3,4]. For example, the clinical outcomes of patients treated with autologous chondrocyte implantation (ACI) are not optimal because many of them form fibrocartilage [2,36]. It is suggested that fibrocartilage formation is due to chondrocyte loss and their function caused by oxidative stress [11]. Therefore, developing new methods to enhance chondrocyte function is required. In this study, we show that UCMSC-EVs maintain cell survival (Figure 3). UCMSC-EV treatment enhanced chondrocyte proliferation and upregulated the mRNA expression of SOX-9, Col-II, and Aggrecan, while downregulating Col-I expression in vitro (Figure 3 and Figure 4). The UCMSC-EVs also reduce oxidative stress in chondrocytes by decreasing mitochondrial superoxide production and enhancing antioxidant enzyme synthesis, including SOD-2 and Sirt-3 (Figure 5). GO enrichment analysis revealed that miRNAs contained within UCMSC-EVs are highly associated with biological processes and molecular functions including small GTPase-mediated signal transduction, GTPase regulatory activity, and components of the mitochondrial matrix (Figure 6 and Figure 7). Moreover, KEGG pathway analysis revealed that miRNAs enriched in UCMSC-EVs may influence chondrocyte function by modulating key signaling pathways, including PI3K-Akt, MAPK, and cAMP (Figure 8). Overall, we found that UCMSC-EVs enhance chondrocyte function, and UCMSC-EV-treated chondrocytes can be used as a potential strategy for chondrocyte-based ACTE.

EVs are generally classified into exosomes, microvesicles, and apoptotic bodies based on their size and biogenetic origin [37]. The International Society for Extracellular Vesicles (ISEV) introduced the Minimal Information for Studies of Extracellular Vesicles (MISEV) guidelines in 2014 to standardize EV characterization practices [38]. These guidelines were subsequently updated in 2018 (MISEV2018) [39] and most recently in 2023 (MISEV2023) [40], reflecting advancements in EV research and analytical methodologies. In accordance with MISEV guidelines, UCMSC-EVs were characterized based on their particle size, protein composition, and morphological features [38,39]. Exosomes are the principal type of EVs widely utilized in cell therapy and tissue engineering applications [3]. In terms of diameter, exosomes range in size from 30 to 100 nm [41]. Through flow cytometry analysis, we discovered that the mean particle sizes of the UCMSC-EVs were 79.8 ± 19.05 nm, corresponding to the sizes of exosomes. According to the guidelines of ISEV, analysis of EVs’ protein composition should include at least three positive markers, with a minimum of one transmembrane or lipid-bound protein—such as CD9, CD63, or CD81 [38,39]. Additionally, the use of at least one negative marker, such as α-tubulin, is recommended to ensure specificity and exclude potential cellular contamination [38,39]. In the present study, isolated UCMSC-EVs tested positive for three transmembrane/lipid-bound protein markers—CD9, CD63, and CD81—consistent with the established criteria for EVs’ identification outlined by the MISEV guidelines (Figure 1C,D). We also discovered that α-tubulin was not detected in our isolated UCMSC-EVs (Figure 1D). The ISEV uses EV as a generic term for particles that are released from cells, are delimited by a lipid bilayer, and cannot replicate on their own (i.e., do not contain a functional nucleus) [39,40]. TEM revealed that the UCMSC-EVs were spherical (Figure 1E). Collectively, the results confirm the successful isolation of UCMSC-EVs, as evidenced by their characteristic size distribution, morphology, and expression of canonical EV surface markers.

Studies have shown that EVs play a vital role in intercellular communication [42,43]. EVs can act directly and bind to specific cells [44]; they also act as paracrine signaling factors that can alter the functions of a neighboring or distant target cell [45]. To assess the effects of UCMSC-EVs on chondrocytes, we examined their uptake and the resulting cellular responses. Coculturing UCMSC-EVs with chondrocytes revealed EV uptake by chondrocytes from day 3 to 7 (Figure 2). These results suggest that UCMSC–chondrocyte communication occurs via EV delivery to chondrocytes. Chondrocyte sources are limited in vivo due to the sparse distribution of cells within hyaline cartilage [4]. Maintaining cell survival and enhancing chondrocyte proliferation in vitro can improve the efficacy of chondrocyte-based ACTE. However, increased cell death or decreased proliferation is indicative of oxidative stress in chondrocytes [11,46,47,48,49]. In the present study, we investigated whether UCMSC-EVs affect chondrocyte survival and proliferation. Our findings demonstrate that UCMSC-EV treatment maintains cell survival and enhances chondrocyte proliferation (Figure 3). These results suggest that UCMSC-EVs maintain cell survival and enhance chondrocyte proliferation, and these may be beneficial for chondrocyte-based ACTE.

Articular cartilage is composed of chondrocytes and hyaline cartilage, and the extracellular matrix (ECM) is mainly composed of Col-II and Aggrecan [4]. Chondrocyte function is essential to the ECM synthesis of articular cartilage [4]. SOX9 is a critical transcription factor for Col-II and Aggrecan synthesis [14]. Fibrocartilage differs from articular cartilage in the ratio of Col-I to Col-II. Hyaline cartilage is predominantly composed of Col-II, while fibrocartilage has higher Col-I levels [50]. In vitro expansion of chondrocytes resulted in the downregulation of Col-II, Aggrecan, and SOX-9 expression, along with the upregulation of Col-I expression [14]. To determine whether UCMSC-EVs enhance chondrocyte function, we analyzed chondrocytes’ mRNA expressions of chondrogenic genes (SOX-9, Col-II, and Aggrecan). We also tested the mRNA expression of fibrocartilaginous (Col-I) chondrocytes after UCMSC-EVs treatment. We discovered that UCMSC-EV treatment increased the mRNA expression levels of SOX-9, Col-II, and Aggrecan by chondrocytes on day 7 (Figure 4). The Col-I expression by the chondrocyte was also decreased after UCMSC-EVs treatment (Figure 4). The results revealed that UCMSC-EVs upregulate chondrogenic and downregulate fibrocartilaginous gene expression in chondrocytes. These results suggest that UCMSC-EVs enhance chondrocyte function.

The imbalance of ROS production and antioxidant defense leads to oxidative stress. Oxidative stress is responsible for the loss of the chondrogenic phenotype in chondrocytes after cell expansion in vitro [14]. Mitochondria are major sources of ROS within cells [16,51]. While chondrocytes primarily rely on glycolysis in articular cartilage, cell expansion in vitro increases mitochondrial oxidative phosphorylation, elevating mitochondrial ROS production and oxidative stress [52,53]. Upregulation of Sirt-3 is indicated to improve oxidative stress by reducing mitochondrial ROS production and enhancing antioxidant defense [54]. Mitochondrial SOD-2 is an antioxidant enzyme and plays a crucial role in controlling ROS production. Superoxide is a proximal mitochondrial ROS produced at respiratory chain complexes I and III and is dismutated by SOD-2 for antioxidant defenses [16,51]. We showed that UCMSC-EV treatment in chondrocytes decreased the level of mitochondrial ROS (Figure 5) and increased the protein levels of SOD-2 and Sirt-3 (Figure 5). These results suggest that UCMSC-EVs enhance chondrocyte function by reducing oxidative stress in chondrocytes.

EVs regulate cell function by transferring lipids, nucleic acids, and proteins to recipient cells [42,43]. miRNAs are small, noncoding RNAs that suppress gene expression at the post-transcriptional level [55], and EVs modulate recipient cells by delivering these regulatory miRNAs [56]. The GO analysis indicated that small GTPase-mediated signal transduction, GTPase regulatory activity, and the mitochondrial matrix were enriched pathways after miRNA analysis in UCMSC-EVs (Figure 7). Previous studies have indicated that ROS produced from small GTPases can regulate mitochondrial function, affecting metabolism and ROS signaling within the cell [57,58,59]. The KEGG pathway analysis revealed that the PI3K-Akt, MAPK, and cAMP signaling pathways may make contributions (Figure 8). The PI3K-Akt and MAPK signaling pathways are intricately involved in the regulation of cellular processes through their interaction with ROS, influencing key functions such as cell survival, mitochondrial morphogenesis, and cellular proliferation [60]. The cAMP signaling pathway affects both mitochondrial functions and ROS production [61]. Elevated mitochondrial cAMP levels have been shown to upregulate Sirt3 expression, thereby conferring cellular protection against ROS-mediated apoptotic pathways [61]. These findings underscore the potential role of miRNAs carried by UCMSC-EVs in enhancing chondrocyte function. Our results suggest that UCMSC-EVs enhance the chondrocyte function through miRNA regulation.

Despite demonstrating enhanced chondrocyte function following UCMSC-EVs treatment in vitro, this study presents several limitations. First, the absence of in vivo validation restricts the ability to assess the therapeutic efficacy, biodistribution, and long-term safety of UCMSC-EVs-treated chondrocytes within articular cartilage defects. Additional animal studies are necessary to bridge the gap between in vitro observations and clinical relevance. Second, the use of UCMSC-EVs derived from a single donor may introduce donor-specific bias, as the composition and functional properties of UCMSC-EVs can vary depending on donor age, health status, and tissue origin. The inclusion of EVs from multiple donors would enhance the generalizability of the findings and account for biological variability. Third, UCMSC-EV dose optimization remains an unresolved issue. The current study employed a fixed EV concentration (10^8^–10^10^ particles/mL) without determining the minimal effective dose or exploring dose–response relationships. Future investigations should aim to establish optimal dosing parameters to improve the therapeutic predictability and safety of UCMSC-EV-treated chondrocytes in chondrocyte-based ACTE applications. Finally, this study profiled only the 50 most abundant miRNAs in UCMSC-EVs, without examining downregulated miRNAs or their specific gene targets in chondrocytes. The KEGG pathways identified—such as PI3K-Akt—are broadly associated with various cell types. Future research should focus on elucidating single-miRNA-driven mechanisms, particularly their influence on mitochondrial function, intracellular signaling pathways, and chondrocyte function, to establish mechanistic insights and enhance clinical applicability.

## 4. Materials and Methods

### 4.1. Materials

Unless stated otherwise, all chemical reagents were obtained from Sigma–Aldrich (St. Louis, MO, USA).

### 4.2. Human Chondrocyte Culture

Human chondrocytes derived from the human articular cartilage were purchased from Lonza Bioscience (cat. no. CC-2550; NHAC-kn Articular Chon CGM, cryo amp; Walkersville, MD, USA). The chondrocytes were cultured and expanded in a monolayer in vitro in Dulbecco’s modified Eagle’s medium (DMEM) (cat. no. BE15-604D; Biowhittaker^®^; Lonza Bioscience, Walkersville, MD, USA) containing 1% nonessential amino acids (NEAA), 1% penicillin/streptomycin (cat. no. 15140122; Gibco BRL, Thermo Fisher Scientific, Waltham, MA, USA), and 10% fetal bovine serum (FBS) [62]. Chondrocytes were maintained at 37 °C in a humidified incubator with 5% CO_2_, and the culture medium was replenished every two days until reaching confluency for passaging [62]. For all of the experiments, the chondrocytes were used within six passages.

### 4.3. Human UCMSC Culture and Isolation of UCMSC-EVs

The human UCMSC-EVs in our study were provided by Kao-An Biomedical. Co., Ltd. (Kaohsiung, Taiwan). For UCMSC-EVs’ isolation, the UCMSCs were isolated from the umbilical cord and cultured in α-MEM medium (cat. no. 12571063; Gibco BRL, Thermo Fisher Scientific, Waltham, MA, USA) containing 5% PLTGold Human Platelet Lysate (cat. no. PLTGold100R; Mill Creek Life Sciences, Inc, Rochester, MN, USA). Every 3 days, the medium was changed and then collected for UCMSC-EVs’ isolation. The UCMSC-EVs were isolated using an ÄKTA™ flux Tangential Flow Filtration system (Cytiva, Life Sciences, Marlborough, MA, USA). Characterization of UCMSC-EVs included assessments of particle size, number, morphological features, and protein composition [39,63]. Isolated UCMSC-EVs were stored at 4 °C and utilized within two weeks to ensure the preservation of bioactivity and structural integrity.

### 4.4. UCMSC-EV Protein Composition and Size Analysis Using Nano Flow Cytometry (nFCM)

For protein composition analysis, the UCMSC-EVs were stained by the following antibodies: isotype control, CD9, CD63, and CD81. The UCMSC-EVs were incubated at 37 °C with fluorophore-conjugated antibodies for subsequent fluorescence-based characterization. After incubation, the mixture was washed with PBS and centrifuged at 100,000× *g* at 4 °C. The pellets were then resuspended in PBS for protein composition and size analysis by nano flow cytometry (nFCM). The UCMSC-EVs were analyzed by nFCM for particle concentration, size distribution, and surface markers analysis. The UCMSC-EVs were diluted to achieve particle counts within the optimal range of 2000–12,000/min. Flow rates and side scattering intensities were converted to the corresponding vesicle concentrations and sizes on NanoFCM software (NanoFCM, Profession V2.330) using calibration curves.

### 4.5. UCMSC-EV Protein Composition Analysis Using Western Blotting

Western blotting was employed to analyze the protein composition of freshly isolated UCMSC-EVs [63]. The EVs were promptly lysed using radioimmunoprecipitation assay buffer (RIPA; Sigma, cat. no. R0278) supplemented with a protease inhibitor cocktail (Biotools, TFU-T25). Protein samples were separated by 12% SDS–PAGE under constant voltage (100 V for 2 h) and subsequently transferred onto a polyvinylidene difluoride (PVDF) membrane (Bio-Rad, cat. no.1620174). Membranes were blocked for 1 h at room temperature with 5% bovine serum albumin in PBS containing 0.1% Tween-20 (PBS-T). Overnight incubation at 4 °C was conducted using primary antibodies against CD81 (Proteintech, Rosemont, IL, USA, 66866-1-lg; 1:1000), CD63, and α-tubulin. Following three washes with PBS-T, membranes were incubated at room temperature for 2–4 h with horseradish peroxidase (HRP)-conjugated secondary antibodies—goat anti-mouse IgG (Proteintech, cat. no. C04001-100) or goat anti-rabbit IgG (Proteintech, cat. no. C04003-100), diluted 1:5000. Protein bands were visualized using SuperSignal West Pico PLUS Chemiluminescent Substrate (Thermo Fisher, cat. no. 34580) and images were captured via the FUSION FX imaging system (Vilber Lourmat, France).

### 4.6. Morphological Characterization of UCMSC-EVs via TEM

UCMSC-EVs’ morphology was examined using TEM [63]. A 10 μL aliquot of freshly isolated UCMSC-EVs suspended in distilled water was applied to a carbon-coated 200-mesh copper grid (Agar Scientific, Rotherham, UK) and dried at 37 °C. The sample was stained with 1% phosphotungstic acid (Scharlau, Barcelona, Spain, cat. no. AC11300100), followed by overnight drying at 37 °C. Imaging was performed using a JEM-1400Plus TEM system (JEOL) operated at 80 kV, and digital micrographs were acquired to visualize the ultrastructural features of the EVs.

### 4.7. UCMSC-EV Treatment of Chondrocytes and Study Groups

The chondrocytes were seeded onto a six-well culture plate at 1 × 10^5^ cells per well and cultured for 24 h in basal medium, which was composed of DMEM supplemented with 1% ascorbic acid, 1% NEAA, 1% penicillin/streptomycin, and 10% FBS. After 24 h, the chondrocytes were treated with UCMSC-EVs. The UCMSC-EV solution was immediately diluted with basal medium before treatment. The chondrocytes were treated with UCMSC-EVs (10^8^ to 10^10^ particles/mL) for 7 days. Two experimental groups were established for this study: (1) a Control group, in which chondrocytes were cultured in basal medium without UCMSC-EVs, and (2) a UCMSC-EV group, in which chondrocytes were cultured in basal medium supplemented with UCMSC-EVs at the indicated concentrations. The culture medium was replaced every two days, and chondrocytes were collected at defined time intervals for subsequent experimental analyses.

### 4.8. Visualization of UCMSC-EV Internalization in Chondrocytes

The UCMSC-EVs at a particle count of 10^10^ were first labeled with ExoSparkler Exosome Membrane Labeling Kit-Red (cat. no. EX02-10; Dojindo Molecular Technologies, Inc. Rockville, MD 20850, USA). Subsequently, red-fluorescence-labeled UCMSC-EVs were isolated through centrifugation per the protocol described in the technical manual of the kit. Chondrocytes were initially seeded onto six-well culture plates and incubated for 24 h. Following medium removal, cells were cultured with red-fluorescence-labeled UCMSC-EVs suspended in basal medium (concentration: 10^9^ particles/mL). The culture medium was refreshed every two days. At designated time points, cells were fixed in 4% paraformaldehyde (in PBS) for 15 min. For cell tracking, chondrocytes were stained with CellTracker™ Green CMFDA (Thermo Scientific™, Waltham, MA, USA, cat. no. C7025), and nuclei were counterstained using DAPI. Internalization of red-labeled UCMSC-EVs by chondrocytes was examined using a confocal laser scanning microscope (Zeiss, Weimar, Germany), and fluorescent images were acquired via camera.

### 4.9. Cell Survival of Chondrocytes After UCMSC-EV Treatment

Chondrocyte viability post-UCMSC-EV treatment was evaluated using a Dr. View live/dead dual-staining kit for mammalian cells (IMT FORMOSA, cat. no. 295; Kaohsiung, Taiwan). Fluorescent images were acquired on days 1 and 5 following treatment. After discarding the medium, cells were rinsed twice with PBS and incubated with the staining solution, which consisted of 0.5 μL calcein-AM (green fluorescence for live cells) and 2 μL ethidium homodimer-1 (EthD-1; red fluorescence for dead cells) diluted in 1 mL of basal medium. The staining process was carried out for 15 min at room temperature. Subsequently, the solution was removed, and chondrocytes were visualized under a fluorescence microscope equipped with excitation filters at 494 nm (calcein-AM) and 528 nm (EthD-1) using the Zeiss ApoTome system (Germany). Cell membrane integrity served as the criterion for assessing cell viability.

### 4.10. Evaluation of Chondrocyte Proliferation Following UCMSC-EV Treatment

Chondrocyte proliferation was assessed using the CellTiter 96^®^ AQueous One Solution Cell Proliferation Assay (Promega, Madison, WI, USA, cat. no. G3582), a colorimetric method based on the mitochondrial reduction of MTS [3-(4,5-dimethylthiazol-2-yl)-2,5-diphenyltetrazolium bromide] into formazan. The amount of formazan generated is directly proportional to the number of metabolically active cells and is quantified by measuring absorbance at 490 nm [64,65,66]. At predetermined time points, an MTS working solution (diluted 1:5 in basal medium) was added to each well containing chondrocytes and incubated for 4 h at 37 °C in a humidified atmosphere (95% air, 5% CO_2_). Subsequently, 100 μL of the medium from each well was transferred to a 96-well plate, and absorbance was recorded using a microplate reader (PathTech, Preston, VIC, Australia) with KC Junior software (KCjunior 1.41).

### 4.11. Detection and Quantification of Mitochondrial Superoxide

Mitochondrial superoxide levels were assessed using MitoSOX™ Red, a fluorescent mitochondrial superoxide indicator (Invitrogen, cat. no. M36008), according to the manufacturer’s protocol. At indicated time points, chondrocytes were washed three times with HBSS and were incubated with 5 μM MitoSOX for 10 min at 37 °C in the dark. After incubation, cells were washed with HBSS and the fluorescence intensity measured by the BioTek Synergy H1 microplate reader with excitation and emission wavelengths of 510 and 580 nm.

### 4.12. Extraction of RNA and Analysis of Gene Expression Using Quantitative Real-Time Polymerase Chain Reaction (qRT-PCR)

At the indicated time points, chondrocytes were harvested for RNA extraction. Total RNA was isolated using the TOOLSmart RNA Extractor (cat. no. DPT-BD24; Biotools, New Taipei City, Taiwan) according to the manufacturer’s protocol. The RNA concentration and purity were assessed by measuring the absorbance ratio at 260/280 nm using a NanoDrop 1000 spectrophotometer (Thermo Fisher Scientific, Waltham, MA, USA). A 260/280 ratio between 1.8 and 2.0 was considered indicative of high-quality RNA without DNA contamination. Subsequently, 0.5–1 μg of total RNA was reverse-transcribed into complementary DNA (cDNA) in a 20 μL reaction volume using the TOOLS Easy Fast RT Kit (cat. no. KRT-BA18; Biotools, New Taipei City, Taiwan), following the manufacturer’s instructions. Quantitative real-time PCR (qRT-PCR) was performed using the TOOLS 2X SYBR qPCR Mix (cat. no. FPT-BB05; Biotools, New Taipei City, Taiwan) on a Bio-Rad CFX detection system (Bio-Rad Laboratories, Hercules, CA, USA). Each reaction contained 2 μL of cDNA template in a total reaction volume of 25 μL. The mRNA levels of interest were quantified using previously published primer sequences for SOX-9, collagen type II (Col-II), collagen type I (Col-I), and glyceraldehyde-3-phosphate dehydrogenase (GAPDH) [67,68,69,70,71,72,73]. The following primer sequences were used: SOX-9 (forward: 5′-CTT CCG CGA CGT GGA CAT-3′; reverse: 5′-GTT GGG CGG CAG GTA CTG-3′); Col-II (forward: 5′-CAA CAC TGC CAA CGT CCA GAT-3′; reverse: 5′-TCT TGC AGT GGT AGG TGA TGT TCT-3′); Col-I (forward: 5′-GGC TCC TGC TCC TCT TAG-3′; reverse: 5′-CAG TTC TTG GTC TCG TCA C-3′); and GAPDH (forward: 5′-TCT CCT CTG ACT TCA ACA GCG AC-3′; reverse: 5′-CCC TGT TGC TGT AGC CAA ATT C-3′). The PCR cycling conditions were as follows: initial denaturation at 94 °C for 1 min, followed by 35 cycles of denaturation at 94 °C for 30 s, and annealing/extension at 59 °C for 30 s. After amplification, a dissociation curve analysis was performed to confirm the specificity of the PCR products. The relative mRNA expression level of each target gene was calculated from the threshold cycle (Ct) value of each PCR product and normalized to GAPDH expression by using the comparative Ct method [74]. For each gene of interest, the readings of four wells from each experimental group were collected at every examined time point.

### 4.13. Antibodies

The antibodies used in this study are as follows: Superoxide Dismutase 2 (SOD-2) (cat. no. 24127-1-AP, proteintech; 1:5000), Sirt-3 (cat. no. 12186-1-AP, proteintech; 1:1000), and β-actin (cat. no. A5441, Sigma).

### 4.14. Western Blot Analysis

At indicated time points, the chondrocytes were washed twice with ice-cold PBS and collected by RIPA buffer (cat. no. ab156034, Abcam, Cambridge, UK) containing a protease inhibitor cocktail (cat. no. TAAR-BBI2, Biotools), and the lysates were cleared by centrifugation at 14,000 rpm for 15 min at 4 °C. The quantified total protein using the BCA protein assay kit (cat. no. 23225, Thermo Scientific™, Waltham, Massachusetts, USA) and 25–35 μg/mL per sample were using for Western blotting and the membranes were blocked in 5% BSA at room temperature for 1 h and then incubated overnight with appropriate primary antibodies. The membranes were then incubated with an HRP-conjugated secondary antibody (goat anti-rabbit IgG (H+L); 1:5000), and goat anti-mouse IgG (Croyez Bioscience Co., Ltd., Taipei, Taiwan; 1:5000) for 2 h at room temperature. The bands were detected by a SuperSignal™ West Pico PLUS Chemiluminescent Substrate (cat. no. 34580, Thermo Scientific™), and band intensity was quantified using a FUSION-FX imaging system (Vilber Lourmat, Collégien, France).

### 4.15. Small RNA Sequencing and Kyoto Encyclopedia of Genes and Genomes (KEGG) Pathway and Gene Ontology (GO) Enrichment Analyses of UCMSC-EVs

#### miRNA Extraction

miRNA was isolated from UCMSC-EVs using the EVmiR Extraction Kit (Topgen Biotech., cat. no. EVmiR-050; Kaohsiung, Taiwan), following the manufacturer’s instructions. The purified miRNA from each sample was eluted in 20 μL of low-TE buffer. Quantification of miRNA concentration was performed using the Qubit microRNA Assay Kit on a Qubit Fluorometer (Thermo Scientific™, Waltham, MA, USA).

### 4.16. miRNA Library Construction for Next-Generation Sequencing

A total of 7 μL of miRNA extracted from UCMSC-EVs was used for small RNA library construction using the VAHTS Small RNA Library Prep Kit for Illumina (Vazyme, cat. no. NR801-01; Nanjing, China) and the VAHTS Small RNA Index Primer Kit for Illumina (Vazyme, cat. no. N816), following the manufacturer’s instructions. The resulting barcoded adapter-ligated amplicons were approximately 142 bp in length. Target bands were excised from 4% agarose gel and purified using the FastPure Gel DNA Extraction Mini Kit (Vazyme, cat. no. DC301-01). Library quality and concentration were assessed by microfluidic electrophoresis using the MultiNA MCE-202 DNA-2500 Kit (Shimadzu, cat. no. 292-27911-91; Kyoto, Japan). The qualified pooled libraries (4 nM) were subjected to single-end sequencing (75 cycles) on an Illumina NextSeq 500 system (San Diego, CA, USA), with all procedures performed according to the manufacturer’s protocols.

### 4.17. Bioinformatics Analysis

Raw sequencing reads were initially processed using CLIP Tool Kit (v1.0.3) to trim adapter sequences at the 3′ end, retaining 34 bp per read [75]. Reads within the insert size range of 15–30 bp, appropriate for mature miRNAs, were selected using Filter FASTQ (v1.1.5) [76]. Filtered reads were aligned to the human miRNA reference (miRBase v22.1) using BWA (v0.7.17.4) [77], and mapping results were assembled and quantified via standard Linux commands. Normalization and differential expression analyses were conducted using edgeR (v3.32.1) [78], identifying miRNAs and genes with log_2_-fold changes >1 or <−1 and a significance threshold of *p* < 0.05.

Target genes of differentially expressed miRNAs were predicted using TargetScan (v7.2), miRanda (accessed 1 April 2025), and miRDB (accessed 1 April 2025). Gene Ontology (GO) and Kyoto Encyclopedia of Genes and Genomes (KEGG) pathway enrichment analyses were performed using ClusterProfiler (v3.18.1) [79], with *p*- and q-values < 0.05 considered significant. Enrichment results were visualized using ggplot2 (v3.3.3) [80].

### 4.18. Statistical Analysis

Experimental data were presented as the mean ± standard error of the mean (SEM), derived from combined replicates. Statistical significance was determined using one-way analysis of variance (ANOVA), followed by post hoc multiple comparisons using Scheffe’s method. A *p*-value less than 0.05 was considered statistically significant.

## 5. Conclusions

We demonstrated that UCMSC-EVs mediate chondrocyte function by enhancing proliferation and upregulating the mRNA expression of SOX-9, Col-II, and Aggrecan, while downregulating Col-I expression. This effect is achieved by reducing oxidative stress in chondrocytes. Our miRNA profiling indicated that UCMSC-EVs influence chondrocyte function through factors including the PI3K-Akt, MAPK, and cAMP signaling pathways. These findings suggest that the effect of UCMSC-EVs on chondrocyte function may be applied for chondrocyte-based ACTE.

## Figures and Tables

**Figure 1 ijms-26-07683-f001:**
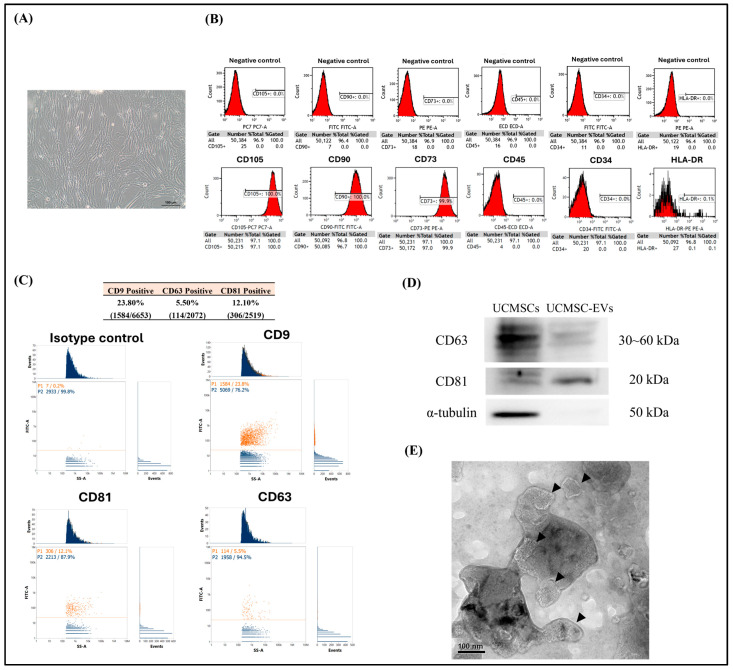
Characterization of UCMSCs and UCMSC-EVs. (**A**) Phase-contrast image of UCMSCs cultured on the plastic culture plate. Scale bar: 100 μm. (**B**) Representative histograms showing the expression of CD105, CD90, CD73, CD45, CD34, and HLA-DR on the surface of UCMSCs. Unstained cells were included as negative controls to validate the specificity of antibody staining. (**C**) Flow cytometry analysis to quantify the percentage of UCMSC-EVs positive for CD9, CD63, and CD81. (**D**) Western blot analysis of the protein expressions of CD63, CD81, and α-tubulin in UCMSCs and UCMSC-EVs. (**E**) Morphology of UCMSC-EVs (arrowhead), as observed using TEM.

**Figure 2 ijms-26-07683-f002:**
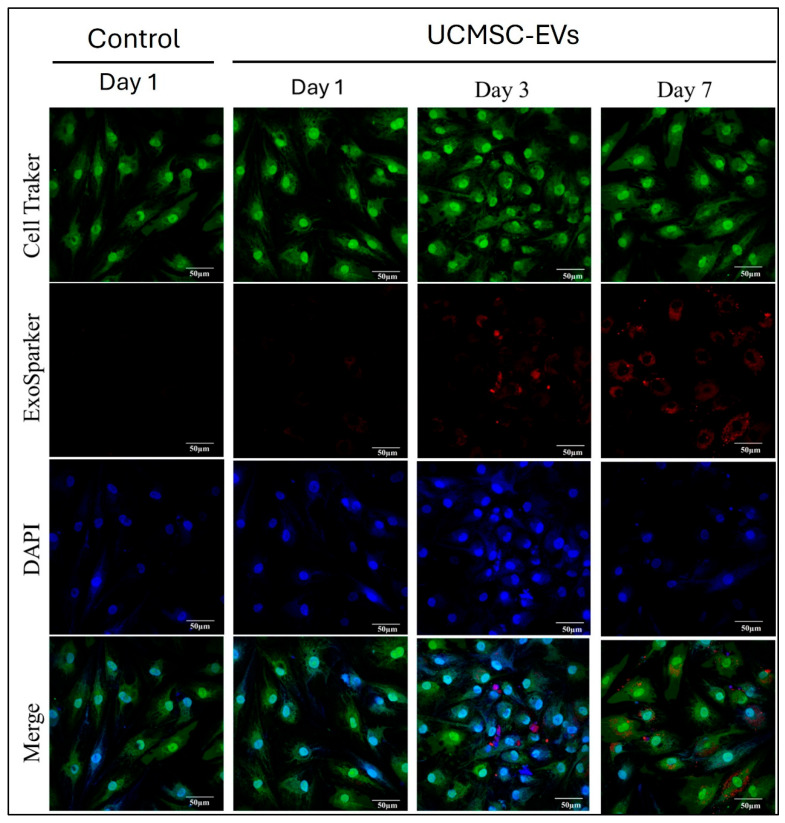
Visualization of UCMSC-EVs internalization in chondrocytes. Chondrocytes were treated with UCMSC-EVs at a concentration of 0 (Control group) or 1 × 10^9^ particles/mL (UCMSC-EVs group) for 7 days. Uptake of UCMSC-EVs by chondrocytes was subsequently evaluated. Uptake of red-fluorescence-labeled UCMSC-EVs in chondrocytes was detected by fluorescence microscopy, and images were obtained using a camera. Green fluorescence stain, cytoplasm; blue fluorescence stain, cell nucleus; red fluorescence stain, ExoSparker-stained UCMSC-EVs. Scale bar: 100 μm.

**Figure 3 ijms-26-07683-f003:**
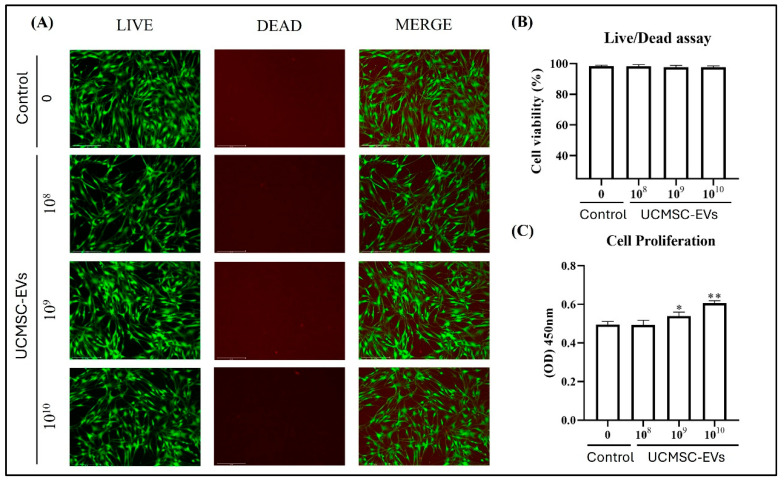
Effect of UCMSC-EVs on survival and proliferation of chondrocytes. Chondrocytes were treated with UCMSC-EVs at concentrations of 0 (Control group) or 10^8^–10^10^ particles/mL (UCMSC-EVs group) for 7 days. (**A**) Live/dead cell staining was performed on day 7 to assess cell viability. Live cells were labeled with calcein-AM (green), while dead cells were stained with ethidium homodimer-1 (red). Green fluorescence indicated viable cells, and red fluorescence indicated non-viable cells. Chondrocytes remained viable following UCMSC-EVs treatment. (**B**) Quantitative analysis of cell survival revealed no detectable cell death after UCMSC-EVs exposure. (**C**) Chondrocyte proliferation was evaluated using an MTS assay on day 7. UCMSC-EVs significantly promoted chondrocyte proliferation, especially at concentrations of 10^9^ and 10^10^ particles/mL. Data are expressed as mean ± SEM (*n* = 6). * *p* < 0.05, ** *p* < 0.01 vs. Control group.

**Figure 4 ijms-26-07683-f004:**
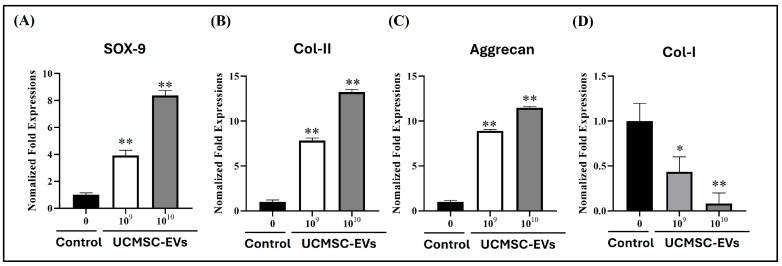
Effect of UCMSC-EVs on mRNA expression of chondrogenic genes (SOX-9, Col-II, and Aggrecan) and fibrocartilaginous gene (Col-I) of chondrocytes. Chondrocytes were cultured for 7 days with UCMSC-EVs at concentrations of 0 (Control group) or 1 × 10^9^ to 1 × 10^10^ particles/mL (UCMSC-EVs group). On day 7, mRNA expression levels of chondrogenic markers ((**A**) SOX-9, (**B**) Col-II, (**C**) Aggrecan) and the fibrocartilaginous marker ((**D**) Col-I) were analyzed. Total RNA was isolated and subjected to real-time polymerase chain reaction analysis. Gene expression levels are presented relative to the Control group, which is normalized to 1. Data are expressed as mean ± standard error of the mean (SEM; *n* = 4). * *p* < 0.05 and ** *p* < 0.01 for comparisons with the Control group.

**Figure 5 ijms-26-07683-f005:**
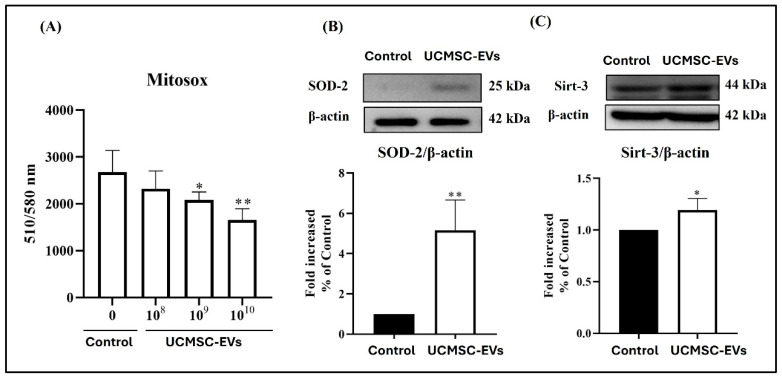
Effect of UCMSC-EVs on oxidative stress of chondrocytes. Chondrocytes were treated with UCMSC-EVs at concentrations of 0 (Control group) or 10^8^–10^10^ particles/mL (UCMSC-EVs group) for 7 days to assess oxidative stress. (**A**) Mitochondrial superoxide levels were evaluated by quantifying MitoSOX Red fluorescence intensity (*n* = 6). (**B**,**C**) Protein expression levels of SOD-2 and Sirt-3 were analyzed by Western blotting following treatment with 10^10^ particles/mL UCMSC-EVs. β-actin was used as the internal control. Protein expression data are normalized to the Control group (defined as 1) and presented as mean ± standard error of the mean (SEM; *n* = 3–5). * and ** indicate *p* < 0.05 and *p* < 0.01, respectively, compared to the chondrocytes in the Control group.

**Figure 6 ijms-26-07683-f006:**
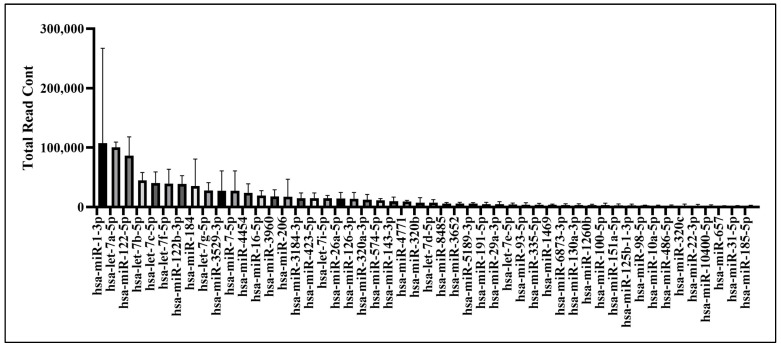
miRNA bioinformatics analysis of UCMSC-EVs. Small RNA sequencing was performed on three independent batches of UCMSC-EVs. The 50 most abundant known miRNAs identified in the UCMSC-EVs were ranked based on total read counts across all replicates (*n* = 3), as shown in the figure.

**Figure 7 ijms-26-07683-f007:**
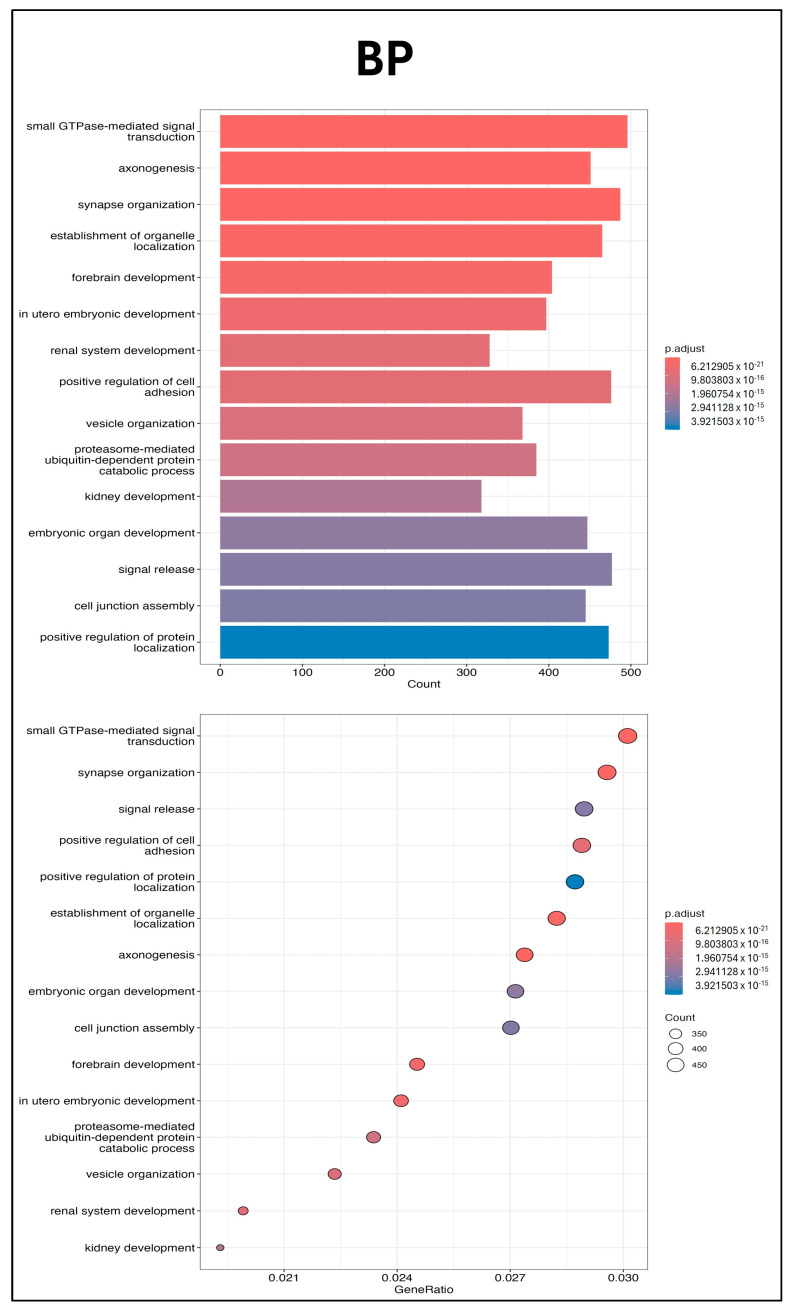
GO enrichment pathway analysis. GO pathway analysis was performed for target genes of miRNAs enriched in UCMSC-EVs. GO biological process (BP), cellular component (CC), and molecular function (MF) terms are presented. Bar and dot plots of GO are shown (*n* = 3).

**Figure 8 ijms-26-07683-f008:**
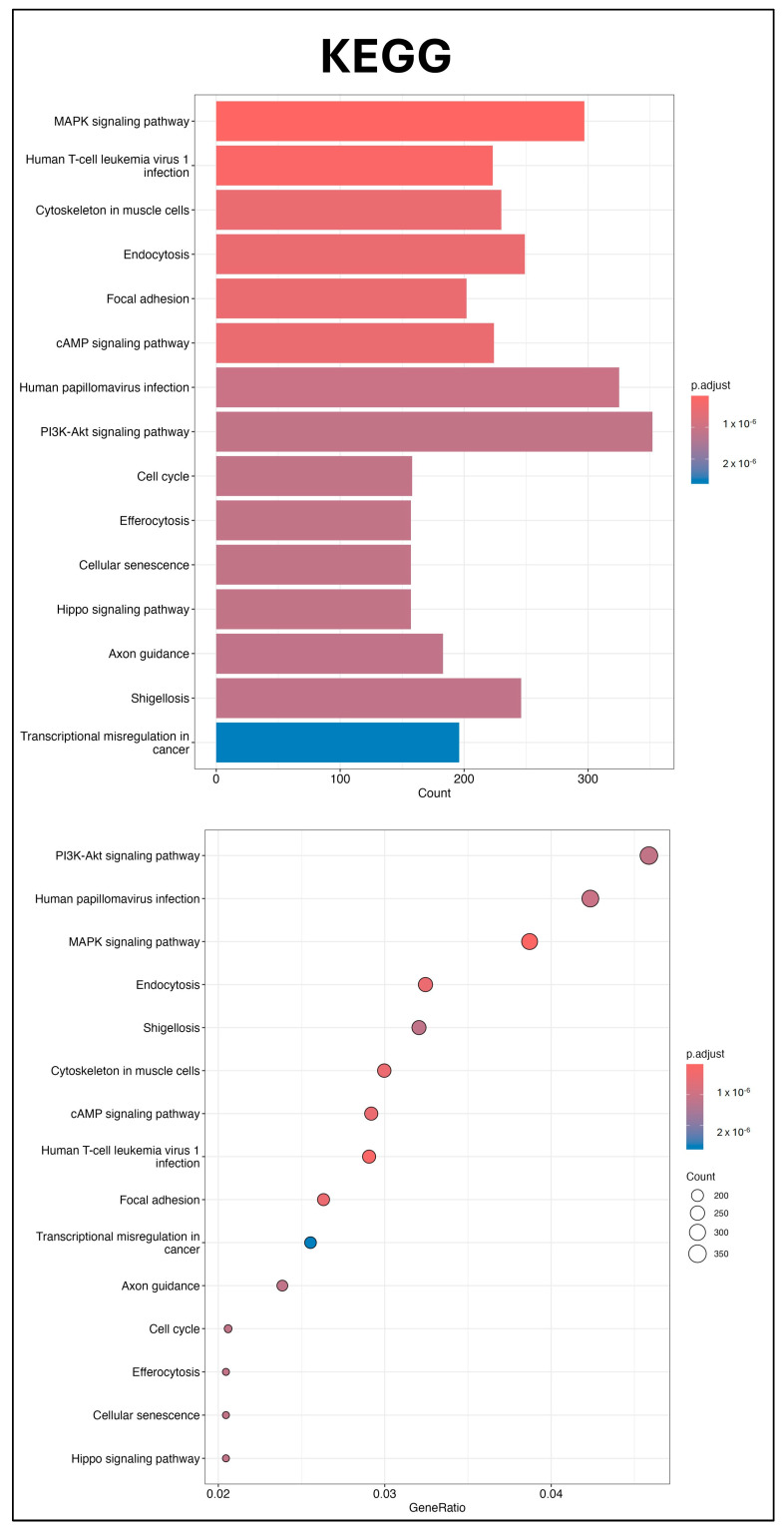
KEGG pathway analysis. KEGG pathway analyses were conducted for predicted target genes of miRNAs enriched in UCMSC-EVs. Representative bar and dot plots summarizing the enriched pathways are presented (*n* = 3).

## Data Availability

The original contributions presented in this study are included in the article. Further inquiries can be directed to the corresponding author.

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
