# Peer review of "Umbilical Cord Mesenchymal Stem Cell-Derived Extracellular Vesicles Enhance Chondrocyte Function by Reducing Oxidative Stress in Chondrocytes"

_ijms, 2025, doi:10.3390/ijms26167683_

Round 1
Reviewer 1 Report
Comments and Suggestions for Authors
- material and methods- all chemicals should have catalog number
- authors should add cell characteristics to confirm phenotype of UCMSCs. Here there is only characterization of EVs from UCMSCs not only UCMSCs.
- why authors did not present reference protein as beta actin or gapdh in western blot for characterization of UCMSC-EVs?
- could you add images of TEM at better quality?
- can you precise isotype control on flow cytometry analysis?
- You showed "that UCMSC-EVs promoted chondrogenic phenotype of chondrocytes" but there is only results at mRNA level. Relative expression on genes was change not phenotype. You should add IF analysis or WB
- Figure 7 and 8 should be better quality
- "We demonstrated that UCMSC-EVs mediate chondrocyte function by enhancing 557
proliferation and chondrogenic phenotype. " authors based on mRNA level, phenotype is not verify. - I cannot verify the result from miRNA because thery are bad quality.
Author Response
- Material and methods- all chemicals should have catalog number.
We sincerely appreciate your expert recommendations. In response, all chemical reagents are now listed with their respective catalog numbers. These changes are reflected in the revised manuscript, specifically in the following section: Results (lines 129, 131, 133, 141, 142-143, 164-165, 168, 176, 184, 201, 209-210, 216, 229, 245, 254, 259, 261, 301, 308-309, 312-313, 315). Modifications are highlighted in red for clarity.
- Authors should add cell characteristics to confirm phenotype of UCMSCs. Here there is only characterization of EVs from UCMSCs not only UCMSCs.
We sincerely appreciate your expert recommendations. In response, the characterization of UCMSCs has now been added into the manuscript. These changes are reflected in the revised manuscript, specifically in the following sections: Results (lines 342-–359), Figure 1, and figure legends (lines 361-366). Modifications are highlighted in red for clarity.
- Why authors did not present reference protein as beta actin or gapdh in western blot for characterization of UCMSC-EVs?
We sincerely appreciate your expert recommendations. In addition to α-tubulin, β-actin was also assessed by western blot for UCMSC-EV characterization. No β-actin signal was detected in the UCMSC-EV samples.
- Could you add images of TEM at better quality?
We sincerely appreciate your expert recommendations. In response, we have replaced the TEM image in figure 1e with improved one. These changes can be found in the revised manuscript (Figure 1e and figure legend, line 366). Modifications are highlighted in red for clarity.
- Can you precise isotype control on flow cytometry analysis?
We sincerely appreciate your expert recommendations. In response, we have revised the manuscript to precise the function of isotype control on flow cytometry analysis. We added the following sentences to the Results section: 'The protein composition of UCMSC-EVs was determined through flow cytometry analysis. An isotype control was used to assess non-specific antibody binding and to differentiate true signals from background noise. The results indicated that UCMSC-EVs were positive for CD9, CD63, and CD81 (Fig. 1C). These changes can be found in the revised manuscript (Result section, lines 348-352). Modifications are highlighted in red for clarity.
- You showed "that UCMSC-EVs promoted chondrogenic phenotype of chondrocytes" but there is only results at mRNA level. Relative expression on genes was change not phenotype. You should add IF analysis or WB.
We sincerely appreciate your expert recommendations. In response, we have revised the phrase 'chondrogenic phenotype' to 'upregulate mRNA levels of SOX-9, collagen type II (Col-II), and aggrecan, while decreasing collagen type I (Col-I) levels. These changes can be found in the revised manuscript (Abstract section, lines 36-38; Introduction section, line 120; Result section, lines 408-411; Figure legend section, lines 423-426; Discussion section, lines 489-491, 543-545, 551-553). Modifications are highlighted in red for clarity.
- Figure 7 and 8 should be better quality
We sincerely appreciate your expert recommendations. Figures 7 and 8 have been re-edited to improve image quality. Both figures were enlarged to achieve higher resolution.
- "We demonstrated that UCMSC-EVs mediate chondrocyte function by enhancing proliferation and chondrogenic phenotype. " Authors based on mRNA level, phenotype is not verify.
We sincerely appreciate your expert recommendations. In response, we have revised the phrase 'chondrogenic phenotype' to 'upregulate mRNA levels of SOX-9, collagen type II (Col-II), and aggrecan, while decreasing collagen type I (Col-I) levels. These changes can be found in the revised manuscript (Abstract section, lines 36-38; Introduction section, line 120; Result section, lines 408-411; Figure legend section, lines 423-426; Discussion section, lines 489-491, 543-545, 551-553). Modifications are highlighted in red for clarity.
- I cannot verify the result from miRNA because they are bad quality.
We sincerely appreciate your expert recommendations. In response, figures 7 and 8 have been re-edited to improve image quality. Both figures were enlarged to achieve higher resolution.

Reviewer 2 Report
Comments and Suggestions for Authors
- Scientific Rationale and Novelty
- While the introduction presents a compelling rationale, the novelty could probably be emphasized a little more clearly – given the many previous studies, how does this study add to or improve upon showing MSC-EV effects on chondrocytes?
- A clearer gap statement would help to justify the focus on UCMSC-EVs over other MSC sources.
- Data Interpretation
- The conclusion that UCMSC-EVs “enhance chondrocyte function” is supported by the data; however, the term is broad. Be specific: enhancement of proliferation, SOX-9/Col-II expression, and reduction of ROS are the actual functional improvements observed.
- Some findings (e.g., non-effect on survival) are mentioned but not deeply discussed - why might survival not be affected while proliferation is?
- Clinical Translation and Limitations
- Consider including into the discussion the below key translational hurdles:
- Scalability and reproducibility of EV isolation.
- Regulatory barriers (EV therapy is still experimental).
- Delivery method into cartilage defects.
- The absence of limitations suggest including a discussion on the lack of in-vivo validation, donor variability (only one UCMSC-EV donor source noted), and EV dose optimization.
Several grammatical issues and awkward phrasings throughout (e.g., “uptake of by cells”, “chondrocyte was found”, “fluorescence-labeled” spelled inconsistently).
Author Response
Reviewer 2:
- Scientific Rationale and Novelty
While the introduction presents a compelling rationale, the novelty could probably be emphasized a little more clearly – given the many previous studies, how does this study add to or improve upon showing MSC-EV effects on chondrocytes?
We sincerely appreciate your expert recommendations. In response, we have added additional phrases to the Introduction section to emphasize the novelty of this study. These include: 'Therefore, UCMSC-EVs may serve as a therapeutic adjunct for restoring chondrocyte function in chondrocyte-based ACTE. Furthermore, despite evidence from in vivo studies supporting the therapeutic potential of UCMSC-EVs, there remains a lack of focused cellular-level investigations, particularly concerning their direct interactions with chondrocyte function. These changes are reflected in the revised manuscript, specifically in the following sections: Introduction (line 111-116). Modifications are highlighted in red for clarity.
A clearer gap statement would help to justify the focus on UCMSC-EVs over other MSC sources.
We sincerely appreciate your expert recommendations. In response, we have added additional sentences to emphasize the advantages of using UCMSCs as a cell source over other MSC sources. These changes are reflected in the revised manuscript (Introduction section, lines 86–87, 89–91 and 95–96). Modifications are highlighted in red for clarity.
- Data Interpretation
The conclusion that UCMSC-EVs “enhance chondrocyte function” is supported by the data; however, the term is broad. Be specific: enhancement of proliferation, SOX-9/Col-II expression, and reduction of ROS are the actual functional improvements observed. Some findings (e.g., non-effect on survival) are mentioned but not deeply discussed - why might survival not be affected while proliferation is?
We sincerely appreciate your expert recommendations. Maintaining cell survival and enhancing chondrocyte proliferation in vitro can improve the efficacy of chondrocyte-based ACTE. However, increased cell death or decreased proliferation is indicative of oxidative stress in chondrocyte. Based on these reasons, we tested the cell survival and proliferation in chondrocytes after UCMSC-EVs treatment. Our findings demonstrated that UCMSC-EV treatment maintains cell survival and enhances chondrocyte proliferation. In response, we have revised phrases to the discussion section to emphasize the effect of UCMSC-EVs on survival and proliferation of chondrocytes. These changes are reflected in the revised manuscript, specifically in the following sections: discussion (line 528-537). Modifications are highlighted in red for clarity.
- Clinical Translation and Limitations
Consider including into the discussion the below key translational hurdles:
Scalability and reproducibility of EV isolation.
Regulatory barriers (EV therapy is still experimental).
Delivery method into cartilage defects.
The absence of limitations suggest including a discussion on the lack of in-vivo validation, donor variability (only one UCMSC-EV donor source noted), and EV dose optimization.
We appreciate the reviewer’s insightful comment regarding the need to address study limitations. In response, we have added a paragraph in the Discussion section outlining key limitations, including the lack of in vivo validation, the use of a single UCMSC-EV donor source, and the absence of EV dose optimization strategies. This addition aims to provide a balanced interpretation of our findings and highlight areas for future investigation. These changes are reflected in the revised manuscript, specifically in the following sections: discussion (line 584-604). Modifications are highlighted in red for clarity.

Reviewer 3 Report
Comments and Suggestions for Authors
This study evaluates the effects of extracellular vesicles (EVs) derived from umbilical cord mesenchymal stem cells (UCMSC-EVs) on chondrocyte function, with particular attention to oxidative stress. The manuscript presents compelling and novel data on the potential of UCMSC-EVs to improve chondrocyte function, with a well-founded rationale and accurate methodology. However, the lack of in vivo validation and comparative comparisons limits the immediate translational impact. It is an excellent preclinical study, but needs further investigation for pubblication:
The efficacy of UCMSC-EVs is not compared to EVs from adult MSCs or other cell types. It is unclear whether UCMSCs are truly superior or simply equivalent.
The ranges (10⁸–10¹⁰ particles/ml) seem arbitrary, without pharmacological justification or based on preclinical data. A systematic dose-response analysis is lacking.
The miRNA profile is only described in the 50 most abundant; downregulated miRNAs or effects on specific gene targets in chondrocytes are not analyzed. The identified KEGG pathways are generic and common to many cell types (e.g. PI3K-Akt), so the specific impact on the chondrocyte phenotype is hypothetical.
Although phenotypic improvements are observed in chondrocytes, cartilage matrix production or hyaline-like cartilage formation has not been assessed.
Author Response
Reviewer 3:
- Several grammatical issues and awkward phrasings throughout (e.g., “uptake of by cells”, “chondrocyte was found”, “fluorescence-labeled” spelled inconsistently).
We sincerely appreciate your expert recommendations. In response, the English in the manuscript has been carefully reviewed sentence by sentence.
- This study evaluates the effects of extracellular vesicles (EVs) derived from umbilical cord mesenchymal stem cells (UCMSC-EVs) on chondrocyte function, with particular attention to oxidative stress. The manuscript presents compelling and novel data on the potential of UCMSC-EVs to improve chondrocyte function, with a well-founded rationale and accurate methodology. However, the lack of in vivo validation and comparative comparisons limits the immediate translational impact. It is an excellent preclinical study, but needs further investigation for publication:
We appreciate the reviewer’s insightful comment regarding the need to address study limitations. In response, we have added a paragraph in the Discussion section outlining key limitations, including the lack of in vivo validation, the use of a single UCMSC-EV donor source, and the absence of EV dose optimization strategies. This addition aims to provide a balanced interpretation of our findings and highlight areas for future investigation. These changes are reflected in the revised manuscript, specifically in the following sections: discussion (line 584-604). Modifications are highlighted in red for clarity.
- The efficacy of UCMSC-EVs is not compared to EVs from adult MSCs or other cell types. It is unclear whether UCMSCs are truly superior or simply equivalent.
We sincerely appreciate your expert recommendations. In response, we have added additional sentences to emphasize the advantages of using UCMSCs as a cell source over other MSC sources. These changes are reflected in the revised manuscript (Introduction section, lines 86–87 and 89–91).
- The ranges (10⁸–10¹⁰ particles/ml) seem arbitrary, without pharmacological justification or based on preclinical data. A systematic dose-response analysis is lacking.
We appreciate the reviewer’s insightful comment regarding the need to address study limitations. In response, we have added a paragraph in the Discussion section outlining this limitation. This addition aims to provide a balanced interpretation of our findings and highlight areas for future investigation. These changes are reflected in the revised manuscript, specifically in the following sections: discussion (line 593-598). Modifications are highlighted in red for clarity.
- The miRNA profile is only described in the 50 most abundant; downregulated miRNAs or effects on specific gene targets in chondrocytes are not analyzed. The identified KEGG pathways are generic and common to many cell types (e.g. PI3K-Akt), so the specific impact on the chondrocyte phenotype is hypothetical.
We appreciate the reviewer’s insightful comment regarding the need to address study limitations. In response, we have added a paragraph in the Discussion section outlining this limitation. This addition aims to provide a balanced interpretation of our findings and highlight areas for future investigation. These changes are reflected in the revised manuscript, specifically in the following sections: discussion (line 598-604). Modifications are highlighted in red for clarity.
- Although phenotypic improvements are observed in chondrocytes, cartilage matrix production or hyaline-like cartilage formation has not been assessed.
We sincerely appreciate your expert recommendations. In response, we have revised the phrase 'chondrogenic phenotype' to 'upregulate mRNA levels of SOX-9, collagen type II (Col-II), and aggrecan, while decreasing collagen type I (Col-I) levels. These changes can be found in the revised manuscript (Abstract section, lines 36-38; Introduction section, line 120; Result section, lines 408-411; Figure legend section, lines 423-426; Discussion section, lines 489-491, 543-545, 551-553). Modifications are highlighted in red for clarity.

Round 2
Reviewer 1 Report
Comments and Suggestions for Authors
Corrections were made however:
- Category number? or catalog number?
- Authors did not present results of unstained cells or isotype control cells at figure named" Flow cytometry analysis to confirm the 363
presence of CD105, CD90, CD73, CD45, CD34 and HLA-DR on the cell surface of UCMSCs."
Author Response
Reviewer 1
- Category number? or catalog number?
We appreciate the reviewer’s helpful comment. We have confirmed that "catalog number" is the appropriate term and have corrected the manuscript accordingly. We have revised the term “category number” to “cat. no.” throughout the manuscript as suggested. These changes are reflected in the revised manuscript, specifically in the following sections: Materials and Methods (line 131, 133, 135, 143, 144, 167, 170, 175, 176,185, 203, 211, 217, 229, 240, 249, 256, 258, 304, 306, 309, 310 ). Modifications are highlighted in red for clarity.
- Authors did not present results of unstained cells or isotype control cells at figure named" Flow cytometry analysis to confirm the presence of CD105, CD90, CD73, CD45, CD34 and HLA-DR on the cell surface of UCMSCs."
We thank the reviewer for this insightful comment. In response, we have added the flow cytometry results of unstained UCMSCs as negative controls to the revised figure. This addition helps validate the specificity of antibody staining and enhances the clarity of marker expression analysis. The updated figure is now included in the revised manuscript as Figure 1. We also revised figure legends in figure 1B. Modifications are highlighted in red for clarity.

Reviewer 3 Report
Comments and Suggestions for Authors
I want to thank the authors for their thorough revision of the manuscript.
Author Response
Reviewer 3
- I want to thank the authors for their thorough revision of the manuscript.
We would like to express our sincere gratitude to the reviewer for their meticulous and insightful review of our manuscript.
